# Machine Learning Based Representative Spatio-Temporal Event Documents Classification

**Byoungwook Kim** [1] , **Yeongwook Yang** [2], **Ji Su Park** [3] **and Hong-Jun Jang** [3,*]

1   Department of Computer Science and Engineering, Gangneung-Wonju National University, Wonju 26403, Republic of Korea
2   Division of Computer Engineering, Hanshin University, Osan 18101, Republic of Korea
3   Department of Computer Science and Engineering, Jeonju University, Jeonju 55069, Republic of Korea
*   Correspondence: hongjunjang@jj.ac.kr; Tel.: +82-63-220-2372

**Abstract:** As the scale of online news and social media expands, attempts to analyze the latest social issues and consumer trends are increasing. Research on detecting spatio-temporal event sentences in text data is being actively conducted. However, a document contains important spatio-temporal events necessary for event analysis, as well as non-critical events for event analysis. It is important to increase the accuracy of event analysis by extracting only the key events necessary for event analysis from among a large number of events. In this study, we define important 'representative spatio-temporal event documents' for the core subject of documents and propose a BiLSTM-based document classification model to classify representative spatio-temporal event documents. We build 10,000 gold-standard training datasets to train the proposed BiLSTM model. The experimental results show that our BiLSTM model improves the F1 score by 2.6% and the accuracy by 4.5% compared to the baseline CNN model.

**Keywords:** BiLSTM; representative spatio-temporal event; document classification





## 1. Introduction

Document classification is the task of assigning documents to categories and is one of the important areas in natural language processing [1]. With the growth of Internet technology and social network services, text data are being created online in large numbers. Text data are produced and distributed through various media such as news, blogs, and social media. As a vast amount of information can be easily obtained, the need for document classification for more efficient document management has increased. Traditionally, document classification problems have been solved using rule-based methodologies, in which rules are defined by humans. However, since there is a problem in that many exception cases occur, machine-learning-based methodologies capable of directly extracting rules from massive data are mainly used recently. Machine learning methods used for document classification are mainly utilized such as Naïve Bayes Algorithm, SVM (Support Vector Machine), Decision Tree, and Artificial Neural Network. Document classification is progressing to an edge computing environment that can classify not only a single server environment but also data stream documents transmitted in real time, and security is enhanced [1].

In the field of natural language processing, event sentence detection is one of the most important problems [2,3]. An event is defined as a specific event that occurs at a specific time and place [4]. Social-media-based data have a much larger and more diverse structure than existing transaction data, but are created, shared, and exchanged through direct mutual communication among members of each society. Therefore, we can discover the general life trends and behavioral patterns of current members of society from social media data. Therefore, studies are being actively conducted to understand the unstructured

and complex data of social media, extract useful knowledge, monitor current major issues, and predict the future.

Existing studies that detect event sentences in text documents focus on accurately detecting sentences with 4W (who, what, when, and where) features based on statistical techniques. That is, all 4W-based events included in the document are detected. However, while there are events in the document that are important for identifying a particular event, there are also events that are not. Insignificant events rather hinder the accurate identification of events. Thus, it is important to find only key events that can represent the contents of a document, rather than all events included in a document. In our previous study, we developed a model to classify representative spatio-temporal documents using CNN [5]. The study used training data consisting only of temporal and spatial attributes, and presented only CNN as a deep learning model. In this study, 'what' and 'who' attributes were added by expanding to the spatio-temporal event perspective. Mislabeled data were corrected while adding attributes to the existing training data. We increased the number of training data from 7400 to 10,000. Among various events that exist in a document, we define an event that is important to the main content of a document as a 'representative spatio-temporal event'. A document containing representative spatio-temporal events is defined as 'representative spatio-temporal event document (RSTEDoc)'. Recurrent neural networks are deep learning models suitable for sequential or variable-length data. LSTM is a model that can maintain long-term dependence between sequence components by solving the vanishing gradient problem that appears in recurrent neural networks. We present the BiLSTM model, which is attracting attention as a natural language processing model.

We summarize our contributions as follows.

- We defined the RSTEDoc in terms of the most important event among multiple events included in a document. Detailed definitions are given in Section 3.1.
- We built 10,000 gold-standard training datasets for classifying RSTEDocs. Since there are no publicly open training data for classifying RSTEDocs yet, we hand-built training data based on large-volume news articles.
- We developed a BiLSTM-based classifier for RSTEDocs and compared to traditional machine learning, using deep learning-based models (CNN and LSTM) as a baseline.
- The BiLSTM-based model had the highest performance among other machine learning models with an F1 score of 0.631 and an accuracy of 0.835.

The rest of our study is organized as follows. In Section 2, we look at related studies on spatio-temporal event detection and deep-learning-based document classification. In Section 3, we define the research problem, and the machine learning model used in this study is briefly described. In Section 4, we describe the data used in the experiment. Section 5 shows the experimental results. In Section 6, we discuss our research results and conclude our study.

## 2. Related Work

### 2.1. Spatio-Temporal Event Detection

A spatio-temporal event is defined as an event that occurs at a specific time and location of interest to stakeholders [6,7]. Natural disasters, crimes, and business events that occurred at a specific time and location are examples of spatio-temporal events. In particular, since events such as natural disasters or environmental pollution consume a lot of social costs, it is important to detect them in real time in order to respond to them [8]. For this reason, studies are being conducted to detect events in text news that is uploaded online every moment.

Spatio-temporal events are generally detected from text data. Text data can contain information about time and place, and this information can be used to detect spatio-temporal events. For example, news articles or social media posts generally contain information about time and place. To detect events from such data, the time and place are extracted and used to detect events. To detect spatio-temporal events, natural language

processing techniques are used to process text data. Sentence classification, named entity recognition, and semantic analysis can be utilized for this purpose. Moreover, there is research on developing techniques to detect spatio-temporal events from image or video data. For instance, in autonomous driving, cameras and radar are used to detect road situations, and techniques to detect spatio-temporal events based on these data are under study. However, generally, spatio-temporal events are detected from text data, and natural language processing techniques are used to process text data.

Hu et al. [9] presented a new document expression method based on word embedding that improves efficiency and accuracy by reducing dimensions and mitigating sparse semantics compared to TF-IDF to efficiently monitor events presented in online news. Chen et al. [10] trained a classification model to identify event-related text in social media, and proposed a clustering-based approach to detect and track events. The authors jointly learned the similarity metric and low-dimensional representation of events using an attention-based neural network, saved and updated the event representation, and demonstrated its effectiveness in large data sets. Nguyen et al. [11] proposed a model to extract and track events from social data streams in real time.

To solve the problem that social media datasets are difficult to accurately extract events from due to noise and short text length, Ahuja et al. [12] proposed Spatio-Temporal Event Detection (STED), which can detect events using other information such as news articles that use a more accurate and realistic vocabulary. The model is used to track events, related topics, time of occurrence, and geospatial distribution across various data sources such as news and Twitter. By modeling news and Twitter together, this model helps filter out noise from Twitter data, and the geographic coordinates and timestamps used in tweets are useful for understanding the spatial and temporal distribution of events. Shah et al. [13] proposed a novel approach to event localization and ranking that can be applied to Twitter data streams. The proposed approach models the language usage of Twitter by hourly city to create a model that detects the magnitude of unexpected change in language usage. This approach uses a space–time grid structure and methods that traverse time, day, state, city, region, and country to detect anomalies in the language spoken in millions of tweets.

Recently, studies to detect spatio-temporal events based on deep learning are also being conducted. Afyouni et al. [14] proposes a hybrid learning model called Deep-Eware for spatio-temporal social event detection. While previous research in event detection was mainly conducted through rule-based or machine-learning-based methods that were only applicable to text data, Deep-Eware combines a deep learning network based on LSTM that considers both spatial and temporal information with a machine learning algorithm based on Random Forest that utilizes spatial attributes. The experimental results show that Deep-Eware outperforms existing research in terms of event detection accuracy. Additionally, the model that considers both spatial and temporal information performs better than the one that only considers text data. These results indicate that the proposed Deep-Eware model could be useful in the field of event detection.

### 2.2. Deep-Learning-Based Document Classification

The development of Internet technology and the increase in the use of social network services cause an increase in the amount of online text data. The task of classifying large amounts of text documents is becoming increasingly important. Since it is a lot of work for humans to classify documents, research on automatic document classification using machine learning is continuously being conducted.

A representative machine learning technique used for document classification is Gaussian Naive Bayes (GNB) [15,16]. The Naive Bayes algorithm is a probability-based algorithm that classifies data by substituting independent events into Bayes' theorem to calculate the probability that data belong to a specific category under the assumption that each event is an independent event. There is also Support Vector Machine (SVM) as a machine learning technique used for document classification [17,18]. SVM is a supervised learning algorithm used for classification and regression analysis. SVM basically separates

the data into two groups by selecting the hyper-plane that is the farthest away from the data. Random Forest (RF) is also a machine learning technique often used for document classification [19,20]. RF is an ensemble algorithm that trains multiple decision tree models and aggregates the results to make predictions.

These various machine learning algorithms have been used for automatic document classification for a long time. Recently, as deep learning has been attracting attention in the field of artificial intelligence, various deep learning techniques have begun to be studied for document classification tasks. Starting to be introduced into deep learning document classification tasks, CNNs are used for document classification [20,21].

Chang et al. [22] proposes a model called DocBERT, which is based on BERT (Bidirectional Encoder Representations from Transformers). DocBERT demonstrates good performance in document classification tasks, thanks to BERT's ability to understand the relationships between words in a document. This paper shows that DocBERT outperforms other models on multiple datasets.

Beltagy et al. [23] proposes a model called Longformer, which has a Transformer structure similar to BERT but has the ability to process long documents. The paper shows experimental results that demonstrate Longformer's superior performance on various document classification tasks.

In our previous study, we developed a CNN-based representative spatio-temporal document classifier [5]. The study used training data consisting only of temporal and spatial attributes, and presented only CNN as a deep learning model. In this study, 'what' and 'who' attributes were added by expanding to the spatio-temporal event perspective. Mislabeled data were corrected while adding attributes to the existing training data. We increased the number of training data from 7400 to 10,000. We introduce a distinction in terms of spatio-temporal events in Section 3.1. Recurrent neural networks (RNN) are deep learning models suitable for sequential or variable-length data. RNN can suffer from the vanishing gradient problem during backpropagation as the input sequence becomes longer. This problem can become more severe as the sequence length increases, making it difficult to solve the problem of long-term dependency. LSTM is a model that can maintain long-term dependence between sequence components by solving the vanishing gradient problem that appears in recurrent neural networks. We present a BiLSTM model, which is attracting attention as a natural language model.

## 3. Materials and Methods

This section introduces a definition of RSTEDoc and describes deep learning models used for classifying RSTEDocs in this study. We briefly review the features of each model.

### 3.1. Definition

The definition of the spatio-temporal event word added 'who' and 'what' attributes from the definition of spatio-temporal information in previous studies [5]. We assume $D = \{d_1, d_2, \ldots, d_n\}$ be a set of documents. $d_i.label$ indicates whether it is a representative spatio-temporal event document (1 means a RSTEDoc, and 0 means a non-RSTEDoc).

Spatio-temporal event word. A document is a set of consecutive sentences and is denoted as $d_i = \{s_1, s_2, \ldots, s_m\}$. A sentence is a set of consecutive words and is denoted as $s_i = \{w_1, w_2, \ldots, w_l\}$. Among the words included in a document, there are words that hold who, what, where, and when information about a specific important event. For example, $w_i.when = \{\text{'December 25'}\}$ denotes that a specific event occurred on December 25. $w_j.where = \{\text{'Seoul'}\}$ denotes that the place where an event occurred is Seoul. $w_i.who = \{\text{'Tourism Council'}\}$ denotes the subject that caused the event. $w_i.what = \{\text{'Seoul Fall Festival'}\}$ denotes that something happened.

Representative of spatio-temporal event word. Multiple spatio-temporal event words can exist in one document. Some spatio-temporal event words may be important words to understand the core content of the document. Among several spatio-temporal words

included in a document, words that are important for understanding the core contents of a document are defined as 'representative spatio-temporal event words'.

Representative spatio-temporal event document. Among the representative space-time events included in a document, a document that has all the attributes of who, what, where, and when is defined as a 'Representative Spatio-Temporal Event Document (RSTEDoc)'.

### 3.2. Machine Learning Models

### 3.2.1. Convolutional Neural Network

A convolutional neural network (CNN) is a model to prevent loss of spatial information in data input to a neural network, which is mainly used to analyze visual images. CNN is a deep learning model that can effectively process images because it effectively analyzes the characteristics of adjacent data while maintaining the spatial information of the data. CNNs are also used for classification tasks [24–26].

In a previous study, we proposed a representative spatio-temporal document classification model using CNN. The layers of the proposed CNN model are shown in Figure 1. The CNN model consists of one word embedding layer, nine convolution layers (conv1d and maxpooling1d), and three dense layers. Among the parameters of the convolution layer, filters are set to 256, kernel to 7, strides to 3, and pool size to 3. Among the parameters of the two dense layers, unit is set to 100 and activation is set to relu. For binary classification, the unit of the last dense layer is set to 1, and the activation is set to sigmoid. We found that the training of the CNN model was slow at the beginning of training, so the model did not learn when stopping patience was set low. Therefore, we set epochs to 300 and stopping patience to 200 in model training. Dropout rate is 0.6, batch size is 64, and learning rate is 0.00001.

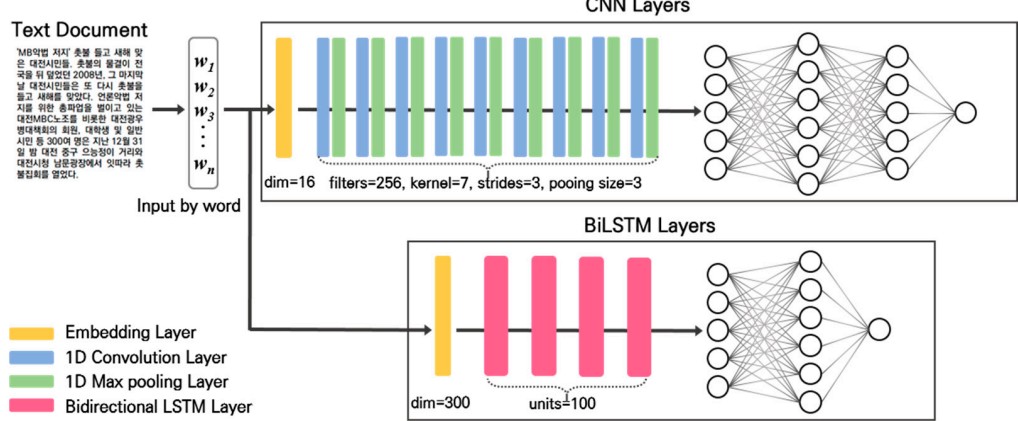

**Figure 1.** The proposed CNN layer and BiLSTM layer-based models.

### 3.2.2. Bidirectional Long Short-Term Memory

The recurrent neural network (RNN) model is a model suitable for analyzing time-series data as it has a recursive structure in which the result of the hidden layer is input to the next hidden layer [27]. However, in the RNN model, as the sequence lengthens, past information is not transferred well to the next step, and as the step is repeated, a loss of gradient problem may occur in which learning weights disappear. LSTM is a model proposed to solve the vanishing gradient problem of RNNs. LSTM distinguishes important information from unnecessary information through the cell state, passes it on to the next state, and can preserve long-term memory even after repeated steps [28]. LSTM is used for text classification [29] and time-series data processing [30].

Although LSTM provides excellent performance, it still has a limitation in that the result tends to converge based on the previous pattern because it still processes the input order in chronological order. In order to solve these disadvantages, BiLSTM (Bidirectional Long Short-Term Memory) has been proposed [31]. BiLSTM shows a structure in which a

forward and a backward hidden layer are connected to the input and output layers, and learning is performed in both directions, and the output value from each hidden layer is calculated to derive the final output value. Since learning is performed in the forward and backward directions separately and then combined, better prediction performance can be expected compared to the existing unidirectional LSTM.

BiLSTM (Bidirectional Long Short-Term Memory) processes input data in both directions, allowing it to utilize both past and future information. This allows for the transition from short-term memory to long-term memory, which is achieved as follows: first, the memory cell state in the forward LSTM is updated using input, forget, and output gates to adjust the memory cell state. Second, the memory cell state in the backward LSTM is updated using input, forget, and output gates to adjust the memory cell state. The output values from the forward and backward LSTMs are concatenated and used as the input for the next layer. This way, BiLSTM adjusts short-term and long-term memory while processing input data in both directions, allowing it to effectively comprehend and process information in input sequences.

The layers of the proposed BiLSTM model are shown in Figure 1. The BiLSTM model consists of a word embedding layer, four Bidirectional LSTM layers, and two dense layers. FastText [32] is used for word embedding in this model. Among the parameters of the Bidirectional LSTM layer, units were set to 100. We implemented a Bidirectional LSTM layer using Keras. Among the parameters of one dense layer, unit is set to 100 and activation is set to relu. For binary classification, the unit of the last dense layer is set to 1, and the activation is set to sigmoid.

To achieve optimal performance in a classification model, it is not enough to apply a BiLSTM layer to the model; one must also find the optimal hyperparameters. Additionally, the depth of the layer that delivers the best performance in terms of time must be identified. We have conducted experiments with various hyperparameters and selected the optimal hyperparameters that demonstrated the best performance.

## 4. Experiment

### 4.1. Datasets

The National Institute of the Korean Language [33] provides various learning data written in Korean text that can be used for artificial intelligence. We use data consisting of a total of 3,536,491 news articles as a corpus. We randomly select 10,000 articles from all articles and build them training data for classifying RSTEDocs. There were 7400 learning data used in the previous study, which consisted only of temporal and spatial attributes. We increased the number of training data to 10,000 for the newly defined RSTEDoc classification and modified the existing training data to fit the new definition. In terms of defining representative spatio-temporal events, 'who' and 'what' were added to the existing data schema. Table 1 shows a sample of the training data used in this study. RSTEDocs have a label attribute of 1, and non-RSTEDocs have a value of 0. 'When' refers to when an event occurred, 'who' refers to who caused the event, and 'where' refers to where the event occurred. In addition, 'what' refers to the title of the event when expressing information about when and what event occurred.

**Table 1.** Example of training data.

| Label | Topic | No. | When | No. | Who | Article |
| | | No. | Where | No. | What | |
|---|---|---|---|---|---|---|
| 0 | life | nan | nan | nan | nan | 'Subway pilgrims' on their way to work … |
| | | nan | nan | nan | nan | |
| 0 | society | nan | nan | nan | nan | Was Congressman Jin Seong-ho … |
| | | nan | nan | nan | nan | |
| 1 | society | 1 | 31 December last year | 1 | citizens | Politicians, please stop fighting … |
| | | 1 | Daegu Sports Park | 1 | New Year's Eve Bell | |
| 1 | society | 2 | Last 31 December | 1 | Daejeon citizens | Citizens of Daejeon celebrate the New Year … |
| | | 2 | Daejeon Jung-gu | 2 | candlelight vigil | |

*4.2. Data Preprocessing*

The three traditional machine learning methods extract feature vectors based on term frequency-inverse document frequency through preprocessing such as tokenization, stop-word removal, and morphological analysis for every sentence in a document. We trained the models by using cross-validation. We used grid search to find the optimal hyperparameters of the three machine learning models.

We segmented the entire document into word units and embedded them. Data composed of about 300 words were the most common, and data of more than 1000 words also existed. We measured the performance by adjusting the size of the data input to the deep learning model from 700 to 1000. If the number of words was less than 700, much of the information contained in the document was lost. When it was set to 700 or more, the change in performance improvement was insignificant, so we set the number of input words to 700.

Intel(R) Xeon CPU with 64 GB memory and GeForce RTX 2080 Ti 11GB GPU were used as a hardware system in this environment. Ubuntu OS 22.04 (64 bit) was utilized as an operating system. CUDA Toolkit was installed for Deep Neural Networks to accelerate the deep learning performance. TensorFlow and Keras were applied as a framework of deep learning. FastText (library for efficient learning of word representations and sentence classification) was used as the word vectorization function for word embedding.

## 5. Result and Discussion

*5.1. Performance Evaluation*

To train and validate the model, we split the training data into training, validation, and test data, as shown in Table 2. RSTEDocs, which are the target data to be predicted in the learning data, account for 31.35% of the total documents.

**Table 2.** Training data distribution ratio for model training.

| Split Data | Count | RSTEDoc | Non-RSTEDoc | Ratio |
|---|---|---|---|---|
| Training | 6000 | 1433 | 4567 | 31.38% |
| Validation | 2000 | 477 | 1523 | 31.32% |
| Test | 2000 | 477 | 1523 | 31.32% |
| Total | 10000 | 2387 | 7613 | 31.35% |

*5.2. Experimental Results*

We compare traditional machine learning methods (GNB, SVM, and RF) and deep-learning-based models (CNN, LSTM, and BiLSTM) to test the classification performance of RSTEDocs. Deep learning basically randomly sets the initial weights of the model.

Therefore, even if the same data are input to the model, the result may be different each time. Therefore, in order to supplement the reliability of the result value due to the randomness of the weight, each experiment was performed 10 times, and the performance was measured with the mean value. Table 3 presents the performance results (precision, recall, F1 score, accuracy, Area Under the ROC Curve (AUC), and Matthews correlation coefficient (MCC) [34]) of each model. We consider accuracy, F1 score, AUC, and MCC as key metrics for evaluating model performance. The ratio of RSTEDocs and non-RSTEDocs is not equal in the training data. Only about 31% of the total data are RSTEDocs. In this case, even if the classifier is not properly trained, the value of precision increases when most of the documents are judged to be non-RSTEDocs, and the value of recall increases when most of the documents are judged to be RSTEDocs.

**Table 3.** Comparison of performance evaluation.

| Machine Learning | Precision | Recall | F1 Score | Accuracy | AUC | MCC |
|---|---|---|---|---|---|---|
| Gaussian Naïve Bayes | 0.587 | 0.278 | 0.348 | 0.767 | 0.794 | 0.362 |
| Linear SVM | 0.643 | 0.453 | 0.526 | 0.789 | 0.812 | 0.383 |
| Random Forest | 0.738 | 0.188 | 0.324 | 0.781 | 0.786 | 0289 |
| CNN | 0.549 | 0.673 | 0.605 | 0.790 | 0.826 | 0.451 |
| LSTM | 0.584 | 0.463 | 0.516 | 0.793 | 0.830 | 0.528 |
| BiLSTM | 0.675 | 0.593 | 0.631 | 0.835 | 0.886 | 0.552 |

The F1 score is 0.632 for BiLSTM, higher than CNN's 0.605. Accuracy is 0.835 for BiLSTM, higher than 0.793 for LSTM. AUC of BiLSTM is 0.886, which was higher than 0.830 of LSTM. MCC also shows the highest value of BiLSTM at 0.552. As shown in Table 3, BiLSTM has the best performance as a model of RSTEDoc classifier. Figure 2 shows the comparison of precision, recall, accuracy, F1 score, AUC and MCC performance for each model.

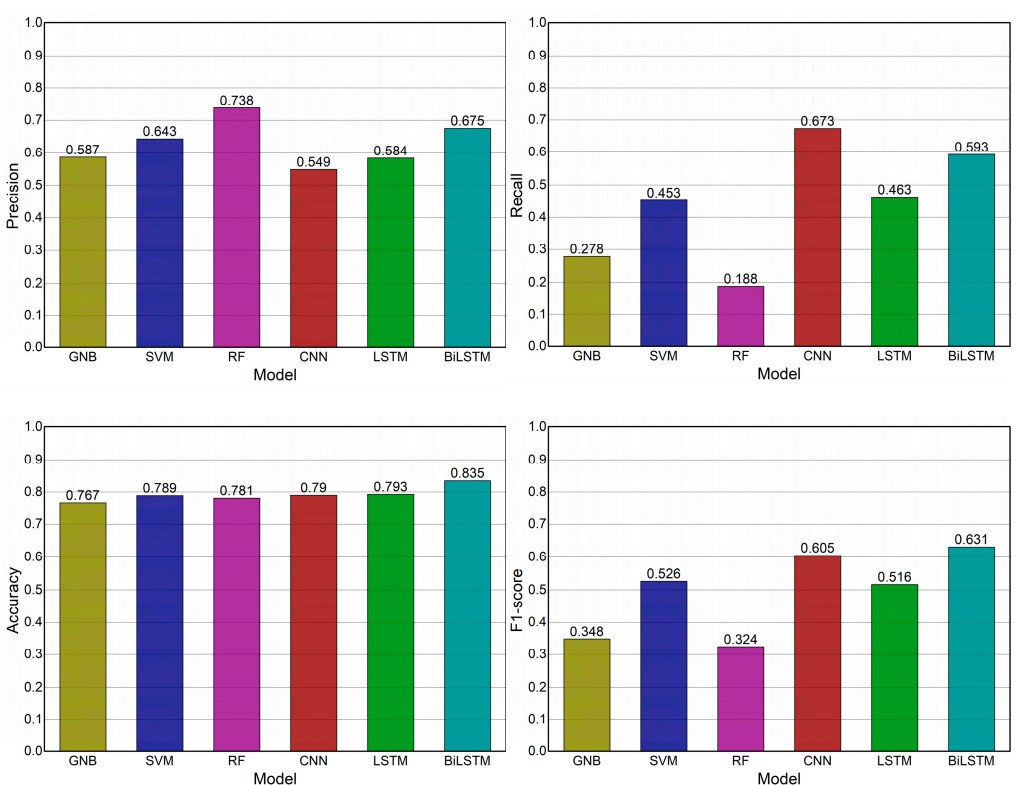

**Figure 2.** *Cont.*

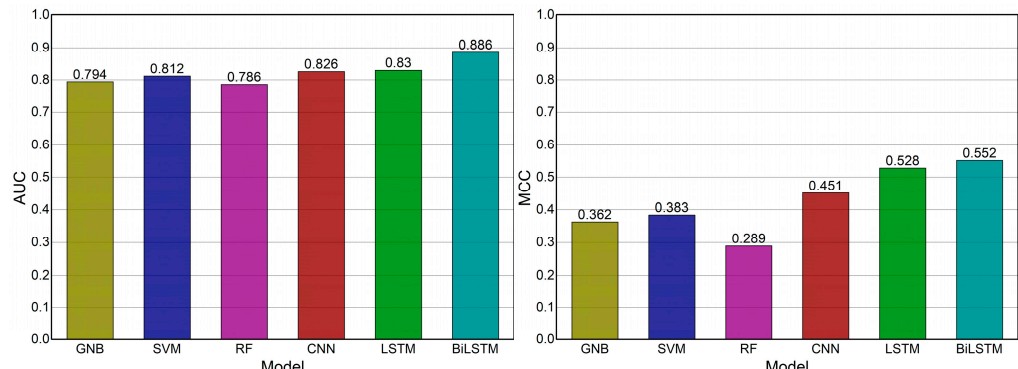

**Figure 2.** Comparison of precision, recall, accuracy, F1 score, AUC, and MCC performance for each model.

AUC is a value that can be a numerical standard in performance evaluation of a binary classification model, and the closer to 1, the closer the graph is to the upper left corner, so it can be a binary classification model with excellent performance.

Figure 3 shows receiver operating characteristic curves for representative spatio-temporal event document classification using CNN, LSTM, and BiLSTM. BiLSTM outperformed CNN and LSTM. CNN and LSTM performed similarly as binary classifiers.

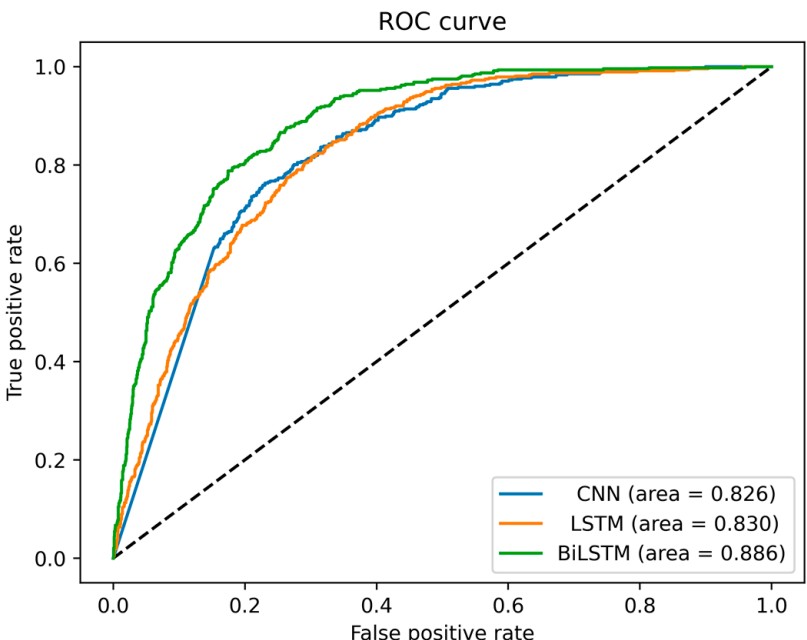

**Figure 3.** Receiver operating characteristic curves of CNN, LSTM, and BiLSTM.

### 5.3. Learning Curve Analysis

To measure the performance of the model according to the training data size (ratio), we plot the learning curves for accuracy and F1 score according to the size of the training data, as shown in Figure 4. CNN and BiLSTM models were trained by dividing the entire data into 20%, 40%, 60%, 80%, and 100%. After 10 experiments were conducted for each model, the average result was derived.

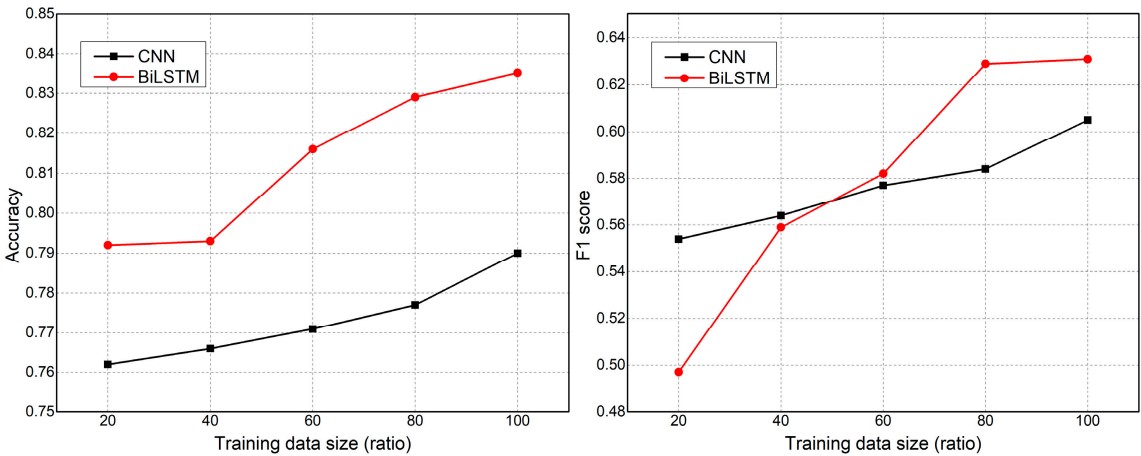

**Figure 4.** Learning curve.

Both the F1 score and accuracy increase linearly as the data size increases. In terms of accuracy, BiLSTM outperforms CNNs at all data sizes while maintaining a constant spacing with CNNs. In terms of F1 score, CNN outperforms BiLSTM when the data size is less than 40%, but BiLSTM outperforms CNN when the data size is 60% or more. In the case of CNN, the F1 score increases gradually from 0.554 to 0.605, whereas in the case of BiLSTM, the F1 score increases relatively rapidly from 0.497 to 0.631. It can be seen that the performance of BiLSTM is more affected by the data size.

## 6. Discussion and Conclusions

In this paper, we propose a BiLSTM-based RSTEDoc classifier. Existing research on event sentence extraction only needs to consider words or grammatical elements related to events. However, classification of RSTEDocs is a difficult problem because even if an event is not important for understanding the contents of a document, it is not classified as a representative spatio-temporal event. We built a gold standard of 10,000 training data to create an RSTEDoc classification model. We conducted an experiment comparing the performance of traditional machine learning methods and deep-learning-based models. Experimental results show that BiLSTM outperforms conventional machine learning classifiers and conventional CNN-based classifiers.

Although BiLSTM outperforms conventional CNNs and LSTMs in RSTEDoc classification performance, the limitation of this study is that the performance is still not high compared to general document classifiers.

Our study does not guarantee the classification performance for other languages or data types outside of Korean news articles, as we constructed our training data using Korean news articles from the National Institute of Korean Language. Therefore, in the future, there is a need to construct training data using various types of documents other than Korean news articles and in languages other than Korean.

Our ultimate research goal is to detect representative spatio-temporal events, and this study is a preliminary investigation into the classification of representative spatio-temporal event documents, which is a necessary step for detecting representative spatio-temporal events. Based on this study, further work is needed to develop a framework for detecting representative spatio-temporal event documents from real-time text data generated online, and apply it in the field.

In order to improve the performance of typical spatio-temporal event document detection, research should be conducted on how to additionally utilize information related to name entity recognition in article text instead of simply using only article text as input data. It is also necessary to conduct research to further refine the model by adding an attention mechanism to the deep learning model in future research.

Recently, deep-learning-based document classification models have demonstrated improved performance by utilizing Graph Convolutional Networks [35–37]. Therefore, as a future research direction, it is necessary to extend the classification model from a simple word-vector-based input to a graph-based model that incorporates information on interactions and relationships between words, sentences, and between sentences, to better utilize the information in the document.

**Author Contributions:** Conceptualization, B.K.; methodology, H.-J.J.; software, B.K.; validation, B.K.; investigation, H.-J.J. and B.K.; data curation, Y.Y.; writing—original draft preparation, B.K.; writing—review and editing, H.-J.J. and J.S.P.; visualization, B.K.; supervision, H.-J.J.; funding acquisition, Y.Y.; project administration, B.K. All authors have read and agreed to the published version of the manuscript.

**Funding:** This research was funded by the Basic Science Research Program through the National Research Foundation of Korea (NRF) funded by the Korean Government (MSIT) (No. 2021R1F1A1049387).

**Institutional Review Board Statement:** Not applicable.

**Informed Consent Statement:** Not applicable.

**Conflicts of Interest:** The authors declare no conflict of interest.

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
