# Peer review of "Machine Learning Based Representative Spatio-Temporal Event Documents Classification"

_applsci, doi:10.3390/app13074230_

Round 1

Reviewer 1 Report

The paper presents an interesting idea but should be modified in some of its parts:

- The related work section should be better organized and fleshed out (it seems a bit lacking in references);

- Figures 1 and 2 are important but should be integrated into a general figure (a sort of graphic abstract) describing the proposed framework. Subsequently, the figure should be used as a reference for the description of the various steps used;

- There are no mentions relating to the software used (design, language, etc);

- In addition to the measures used in table 3, measures more sensitive to data imbalance should be added, such as balanced accuracy or Matthew's correlation coefficient;

- What happens if you organize data in a graph? A recent paper that should be cited talks about it:

Manzo, M., & Pellino, S. (2021). FastGCN+ ARSRGemb: a novel framework for object recognition. Journal of Electronic Imaging30, 033011.

Reviewer 2 Report

1. Show the architecture of your neural network. What new have you proposed? Because  Fig. 2 is unclear.

2. In Section 3.2.2 Bidirectional-Long Short-Term Memory, you create a model. How do you make the transition from short-term memory to long-term memory?

3. How did you get the value 2387 in the "Total" cell in Table 2 in the RSTEDoc column? Perhaps the first cell should not contain 1.433, but 1433? And in the Ratio column, you do not calculate the sum, but the arithmetic average.

4. Rework table 1, because it is difficult to understand what information you put there

5. Fig.3 is not informative at all. Throw away this drawing

Reviewer 3 Report

The paper presents the extension of the previous research of the authors on spatio-temporal event documents classification. The authors compare the performances of traditional machine learning methods and deep learning-based models. The paper is well-written and presents the original experimental results. 

An important drawback is the discussion part. There is a need for some discussion concerning the threats to validity of the research results. Especially, if the results are generalizable to other languages or types of text data beyond Korean news articles provided by the National Institute of the Korean Language. Moreover, there should be also a discussion concerning the application of the results in practical settings and its potential limitations.

It would be also wise to discuss the directions for future research which stems from the findings of this study.

Round 2

Reviewer 1 Report

No further changes are required

Reviewer 2 Report

no comments

Reviewer 3 Report

The authors have improved the manuscript based on my comments. I do not have further suggestions.